# *Poria cocus* Wolf Extract Ameliorates Hepatic Steatosis through Regulation of Lipid Metabolism, Inhibition of ER Stress, and Activation of Autophagy via AMPK Activation

**DOI:** 10.3390/ijms20194801

**Published:** 2019-09-27

**Authors:** Ji-Hyun Kim, Hyun A Sim, Dae Young Jung, Eun Yeong Lim, Yun Tai Kim, Byung Joo Kim, Myeong Ho Jung

**Affiliations:** 1Healthy Aging Korean Medical Research Center, School of Korean Medicine, Pusan National University, Yangsan 50612, Korea; kimji77@pusan.ac.kr (J.-H.K.); dyjung999@naver.com (D.Y.J.); vision@pusan.ac.kr (B.J.K.); 2Division of Longevity and Biofunctional Medicine, School of Korean Medicine, Pusan National University, Yangsan 50612, Korea; andrea0224@naver.com; 3Division of Functional Food Research, Korea Food Research Institute, Jeollabuk-do 55365, Korea; 50005@kfri.re.kr (E.Y.L.); ytkim@kfri.re.kr (Y.T.K.); 4Department of Food Biotechnology, Korea University of Science & Technology, Daejeon 34113, Korea

**Keywords:** *Poria cocos* Wolf, hepatic steatosis, AMP-activated protein kinase, endoplasmic reticulum stress, autophagy, lipogenesis, fatty acid oxidation

## Abstract

*Poria cocos* Wolf (PCW) is an edible, pharmaceutical mushroom with remarkable biological properties including anti-tumor, anti-inflammation, anti-oxidation, anti-ageing, and anti-diabetic effects. In the current study, we investigated the effects of PCW extract on hepatic steatosis under in vitro and in vivo conditions, and elucidated the underlying mechanisms. In this study, a mixture of HepG2 cells treated with free fatty acid (FFA)—palmitic and oleic acid—and high-fat diet (HFD)-fed obese mice were used; in this background, the triglyceride (TG) levels in HepG2 cells and mice liver were measured, and the expression levels of genes associated with lipogenesis, fatty acid oxidation, endoplasmic reticulum (ER) stress, and autophagy were determined. Treatment of HepG2 cells with FFA enhanced intracellular TG levels in HepG2 cells, but co-treatment with PCW significantly attenuated the TG levels. Notably, PCW significantly enhanced the phosphorylation of AMP-activated protein kinase (AMPK), acetyl-CoA carboxylase (ACC), and sterol regulatory element-binding protein-1c (SREBP-1c) in FFA-treated HepG2 cells. PCW downregulated the expression of lipogenesis-related genes, but upregulated the expression of genes associated with fatty acid oxidation. Further, PCW inhibited FFA-induced expression of ER stress markers and induced autophagy proteins. However, inhibition of AMPK significantly attenuated the beneficial effects of PCW in HepG2 cells. Moreover, PCW efficiently decreased HFD-induced hepatic TG accumulation in vivo and increased the phosphorylation of hepatic AMPK. Three compounds present in PCW including poricoic acid, pachymic acid, and ergosterol, significantly decreased FFA-induced increase in intracellular TG levels, consistent with increased AMPK phosphorylation, suggesting that poricoic acid, pachymic acid, and ergosterol are responsible for PCW-mediated amelioration of hepatic steatosis. Taken together, these results demonstrated that PCW ameliorates hepatic steatosis through the regulation of lipid metabolism, inhibition of ER stress, and activation of autophagy in an AMPK-dependent manner. This suggested that PCW can be potentially used for the treatment of hepatic steatosis.

## 1. Introduction

Non-alcoholic fatty liver disease (NAFLD) is the most common cause of liver problems, ranging from hepatic steatosis to more severe diseases, including non-alcoholic steatohepatitis (NASH), fibrosis, cirrhosis, and hepatic carcinoma [1]. Hepatic steatosis is the first step in the development of NAFLD, and is characterized by excessive triglyceride (TG) accumulation caused by the following: increased de novo lipogenesis, decreased fatty acid β-oxidation in the liver, export of very-low-density lipoprotein (VLDL) from the liver, and continued lipolysis in the adipocytes [2]. NAFLD leads to the development of many metabolic diseases such as obesity, type-2 diabetes, insulin resistance, and hypertriglyceridemia [1]. Therefore, development of agents capable of alleviating hepatic steatosis may represent a therapeutic approach to the treatment of hepatic disorders, specifically associated with NAFLD. However, up till now, there is no effective and safe therapy against NAFLD except for a few obesity-targeting interventions and life style-mediated weight loss. Many dietary phytochemicals have gained attention for the treatment of NAFLD as the use of phytochemicals is considered an alternative strategy for developing effective and safe drugs against NAFLD [3].

AMP-activated protein kinase (AMPK) is a serine/threonine kinase that plays a critical role in energy homeostasis and nutrient sensing, and is activated by low cellular energy status [4]. AMPK exists as a heterotrimeric complex comprising a catalytic subunit (α) and two regulatory subunits (β and γ) and is expressed in almost all tissues. AMPK is activated by the phosphorylation of the Thr172 residue in the α subunit [4]. The activation of AMPK inhibits ATP-consuming processes including fatty acid synthesis and gluconeogenesis, while it stimulates ATP-generating processes such as fatty acid oxidation and glycolysis [4]. Thus, AMPK is a likely therapeutic target for treating metabolic diseases including obesity, insulin resistance, type-2 diabetes, NAFLD, and cardiovascular disease (CVD).

AMPK activation ameliorates hepatic steatosis through multiple mechanisms [5,6,7]. The activation of AMPK inhibits fatty acid synthesis (lipogenesis), whereas it stimulates fatty acid oxidation. AMPK activation results in the phosphorylation and inactivation of acetyl-CoA carboxylase (ACC), thereby leading to reduced levels of malonyl-CoA, a precursor to fatty acid synthesis, a potent inhibitor of carnitine palmitoyltransferase-1 (CPT-1), and a rate-limiting enzyme in fatty acid oxidation. Thus, the reduction of malonyl-CoA levels by AMPK activation leads to decreased lipogenesis and increased mitochondrial fatty acid oxidation. Furthermore, AMPK activation directly phosphorylates the sterol regulatory element-binding protein 1c (SREBP1c) at Ser372, a master transcription factor of lipogenesis, while inactivating its transcriptional activity resulting in repression of genes involved in lipogenesis, such as fatty acid synthase (*FAS*), stearoyl-coenzyme A desaturase 1 (*SCD1*), and *ACC1*. The anti-lipogenesis activity of AMPK, therefore, makes this enzyme a potential therapeutic target for the treatment of hepatic steatosis.

AMPK inhibits the lipid-induced ER stress and TG accumulation in hepatocytes [6]. Prolonged endoplasmic reticulum (ER) stress conditions in which ER homeostasis is not recovered are characterized by increased expression of key markers, such as glucose-regulated protein 78 (GRP78), C/EBP homologous protein (CHOP), and phosphorylated protein kinase-like ER kinase (p-PERK), which induce hepatic steatosis [8]. ER stress increases hepatic lipogenesis, inhibits the assembly and secretion of VLDL, induces VLDL receptor expression, and promotes insulin resistance [8]. Thus, protection against prolonged ER stress can serve as a good strategy for the treatment of hepatic steatosis.

AMPK suppresses hepatic lipid accumulation by stimulating autophagy, a catabolic process that removes dysfunctional macromolecules and organelles via lysosomal degradation [7]. Autophagy plays a critical role in the hydrolysis of intracellular lipid droplets to maintain lipid homeostasis—a phenomenon commonly referred to as lipophagy [9]. AMPK activation positively regulates autophagy by suppressing the mammalian target of rapamycin (mTOR) signaling, which is an inhibitor of autophagy [10]. Many bioactive compounds purified from medicinal herbs lead to the attenuation of hepatic lipid accumulation via AMPK activation [3]. Therefore, recent research has focused on the activation of autophagy and the development of AMPK activators as promising treatment strategies for hepatic steatosis.

*Poria cocos* Wolf (PCW) is an edible medicinal mushroom that grows on the roots of pine trees and is widely used as herbal medicine in China, Japan, and Korea [11]. PCW has remarkable biological activities including anti-tumor, anti-inflammatory, anti-oxidative, anti-ageing, anti-hepatic, anti-diabetic, and anti-hemorrhagic fever effects [11,12,13]. The major bioactive components in PCW include triterpenoids, fatty acids, sterols, and polysaccharides [11]. These organic compounds present in PCW are used in many traditional Chinese prescriptions to treat gastritis, nephrosis, edema, dizziness, nausea, emesis, and hyperglycemia [11]. However, the pharmaceutical effects of PCW against NAFLD have not been reported so far.

Therefore, in this study, we evaluated the protective effects of PCW extract against hepatic steatosis and characterized the underlying mechanisms in fatty acid-treated HepG2 cells and mice fed a high-fat diet (HFD). Here, to our knowledge, we demonstrated for the first time that PCW ameliorates hepatic steatosis through regulation of lipid metabolism, inhibition of ER stress, and activation of autophagy in an AMPK-dependent manner, which further supports the hypothesis that PCW can be used as a potential treatment for NAFLD.

## 2. Results

### 2.1. Cytotoxicity of Fatty Acid Mixture and PCW on HepG2 Cells

To mimic hepatic steatosis in vitro, HepG2 cells were exposed to free fatty acid (FFA) containing palmitate and oleate (1:2 ratio). First, we examined the cytotoxicity of a mixture of FFA and PCW on HepG2 cells. As shown in Figure 1A,B, cell viability was not affected by independent treatment with 1 mmol/mL FFA and 80 μmol/mL of PCW in HepG2 cells. However, a combinatorial treatment with 1 mmol/mL FFA and 80 μmol/mL PCW exhibited cytotoxicity in HepG2 cells (Figure 1C), although a combination of 1 mmol/mL FFA and 40 μmol/mL PCW exhibited less cytotoxicity (Figure 1C). Therefore, co-treatment of HepG2 cells with FFA and PCW was performed using 1 mmol/mL FFA and 40 μmol/mL PCW in subsequent experiments.

### 2.2. PCW Inhibits FFA-Induced TG Accumulation in HepG2 Cells

We assessed the inhibitory effect of the PCW extracts on FFA-induced intracellular TG accumulation in HepG2 cells. Oil Red O (ORO) staining revealed that compared to the untreated HepG2 (control) cells, FFA-treated cells had higher intracellular TG levels. However, pre-treatment with PCW significantly blocked the FFA-induced TG accumulation (Figure 2A). Enzymatic measurement of intracellular TG content using a commercial kit revealed that PCW substantially decreased the FFA-induced TG content compared to that in HepG2 cells not treated with PCW (Figure 2B). These results indicated that PCW extracts attenuate FFA-induced intracellular TG accumulation in HepG2 cells.

### 2.3. PCW Activates AMPK Pathway in HepG2 Cells Treated with FFA

AMPK activation inhibits hepatic steatosis through multiple pathways [5,6,7]. To elucidate the mechanisms underlying the preventive effect of PCW on FFA-induced TG accumulation in HepG2 cells, we assessed the activation of AMPK and its downstream signaling in HepG2 cells. As shown in Figure 3A, treatment with the PCW extract induced the phosphorylation of AMPK at Thr172, and subsequently, the phosphorylation of ACC at Ser-79—a downstream target of AMPK in HepG2 cells—indicating that PCW activated AMPK directly in HepG2 cells. Furthermore, we evaluated the beneficial effects of PCW on AMPK activation in FFA-treated HepG2 cells. As shown in Figure 3B, treatment with FFA markedly reduced the phosphorylation of AMPK and ACC in HepG2 cells. However, pre-treatment with PCW effectively reversed the phosphorylation of both AMPK and ACC. Taken together, these results demonstrated that PCW activates AMPK pathway in HepG2 cells, which may play an important role in the attenuation of FFA-induced intracellular TG accumulation in HepG2 cells.

### 2.4. PCW Inhibits the Expression of Lipogenesis Genes and Stimulates the Expression of Fatty Acid Oxidationgenes in HepG2 Cells Treated with FFA

AMPK activation phosphorylates SREBP1c—a critical transcription factor for the stimulation of lipogenesis genes, thereby inhibiting transcriptional activity of SREBP1c, which leads to decreased lipogenesis via downregulation of SREBP1c-regulated lipogenesis genes including *FAS*, *SCD1,* and *ACC* as well as *SREBP1c* [5]. Therefore, we investigated the phosphorylation of SREBP1C and the expression of its target genes involved in lipogenesis in FFA-treated HepG2 cells. As shown in Figure 4A, treatment with FFA resulted in a minimally phosphorylated SREBP—an inhibitory SREBP1c form—but increased the protein levels of SREBP1c (both precursor and mature form) and its target lipogenesis gene, *FAS*. However, PCW pre-treatment significantly reversed these FFA-induced effects (Figure 4A). Furthermore, qPCR assay revealed that FFA treatment increased the mRNA levels of SREBP1s and its target lipogenesis genes, including *FAS*, *ACC1*, and *SCD1*; however, PCW pre-treatment reversed FFA-mediated increase in mRNA levels of these lipogenesis genes (Figure 4B). We then confirmed the inhibitory effects of PCW on the transcriptional activity of *SREBP1c* in HepG2 cells on the basis of the activities of SREBP1c and FAS promoters containing the SREBP-response element. As shown in Figure 4C, treatment with FFA stimulated the activities of both SREBP1c and FAS promoters, which was reversed by pre-treatment with PCW. These results suggested that PCW downregulates lipogenesis genes through inhibition of SREBP1c.

AMPK activation stimulates fatty acid oxidation, which can attenuate hepatic steatosis [5]. Therefore, we assessed the expression of fatty acid oxidation-related genes in FFA-treated HepG2 cells. As shown in Figure 5A, FFA treatment resulted in a decrease in both protein and mRNA levels of peroxisome proliferator-activated receptor-α (*PPARα*), a key transcription factor of fatty acid oxidation gene, but PCW pre-treatment reversed this decrease. qPCR data also revealed that FFA treatment resulted in a decrease in the mRNA levels of *PPARα* target genes including carnitine palmitoyltransferase (*CPT1*) and acyl-coenzyme A oxidase (*ACO*) (Figure 5B); however, the PCW extract reversed this decrease in mRNA levels. Taken together, these results demonstrated that PCW inhibits lipogenesis but stimulates fatty acid oxidation, which can lead to reduced intracellular TG accumulation in HepG2 cells.

### 2.5. PCW Alleviates ER Stress in HepG2 Cells Treated with FFA

AMPK activation negatively regulates ER stress, which induces hepatic steatosis through increased lipogenesis and reduced apolipoprotein secretion [6]. We investigated whether PCW-mediated attenuation of TG accumulation was associated with alleviation in ER stress. To this end, we examined ER stress markers including GRP78, CHOP, XBP1c, and p-PERK in FFA-treated HepG2 cells. The western blot revealed that FFA treatment induced GRP78, CHOP, XBP1c, and p-PERK, and increased their protein levels. However, PCW reversed the induction of these proteins, suggesting that PCW is capable of alleviating FFA-induced ER stress (Figure 6). Taken together, these results suggested that ER stress alleviation may play a role in the protection of PCW against hepatic steatosis.

### 2.6. PCW Activates Autophagy in HepG2 Cells Treated with FFA

AMPK activation induces autophagy, which plays an important role in the prevention of hepatic steatosis [9]. In order to understand the effect of PCW in the induction of autophagy, we examined the expression of autophagy markers in FFA-treated HepG2 cells. As shown in Figure 7A, FFA treatment reduced the levels of autophagosome-related proteins, including LC3A/B, coiled-coil, myosin-like BCL2-interacting protein (Beclin) 1, and autophagy-related protein (ATG) 3, 7, and 16, in HepG2 cells. However, pre-treatment with PCW reversed these effects (Figure 7A). Autophagy is regulated by the mTOR/p70S6K pathway that inhibits autophagy through inhibition of autophagosome formation [10]. We therefore investigated whether PCW-mediated activation of autophagy was associated with mTOR/p70S6K in HepG2 cells. As shown in Figure 7B (left), treatment with FFA increased the levels of phosphorylated mTOR and p70S6K, the downstream components of mTOR signaling; however, PCW pre-treatment reversed these effects. Furthermore, treatment with PCW decreased the phosphorylation of mTOR and p70S6K directly (Figure 7B, right), indicating that PCW activates autophagy through the inhibition of mTOR/p70S6K signaling. These results suggested that PCW-induced activation of autophagy may play an important role in the attenuation of hepatic steatosis by PCW.

### 2.7. Pre-Treatment with Compound C, an AMPK Inhibitor, Reverses PCW-Mediated Beneficial Effects on Hepatic Steatosis in HepG2 Cells Treated with FFA

To confirm whether PCW-mediated beneficial effects on hepatic steatosis are associated with AMPK activation, we assessed AMPK activation and the intracellular TG accumulation in FFA-treated HepG2 cells treat pretreated with compound C, an AMPK inhibitor. As shown in Figure 8A, PCW treatment reversed FFA-mediated reduction in p-AMPK and p-ACC levels; however, pre-treatment with compound C inhibited PCW-induced phosphorylation of AMPK and ACC. (Figure 8A). Measurement of intracellular TG content revealed that PCW prevented FFA-induced TG accumulation, while pre-treatment with compound C abrogated this preventive effect (Figure 8B). This suggested that AMPK activation is associated with PCW-mediated attenuation of hepatic steatosis. Furthermore, to prove whether the protective mechanisms of PCW on hepatic steatosis are associated with AMPK activation, we determined the expression of lipid metabolism genes, ER stress markers, and autophagosome formation-related proteins in HepG2 cells pretreated with compound C. Consistent with the previous results, PCW reversed FFA-mediated increase in mRNA levels of lipogenesis genes (Figure 9A) and decrease in mRNA levels of fatty acid oxidation genes (Figure 9B). Pre-treatment with compound C blocked these PCW-mediated effects, and also inhibited PCW-mediated beneficial effects on ER stress markers (Figure 9C) and autophagosome formation-related protein (Figure 9D) in FFA-treated HepG2 cells. Taken together, these results demonstrated that the protective mechanisms of PCW on hepatic steatosis are mediated through AMPK activation.

### 2.8. PCW Protects against HFD-Induced Hepatic Steatosis In Vivo

We evaluated the in vivo effects of PCW on hepatic steatosis in HFD-induced obese mice. Histological examination revealed that HFD feeding resulted in the development of a white-colored fatty liver; however, PCW administration resulted in the maintenance of a relatively healthy liver (Figure 10A). HFD feeding resulted in increased liver weight and higher ratios of liver:body weight, while PCW administration significantly blocked this increase in HFD-fed obese mice (Figure 10B). Measurement of hepatic TG levels revealed that PCW administration led to a decrease in TG levels (Figure 10C). Taken together, these results indicated that PCW ameliorates fatty liver in HFD-induced obese mice. Finally, to confirm whether PCW-mediated amelioration of hepatic steatosis was dependent on the AMPK pathway, we assessed AMPK phosphorylation in the liver of HFD-fed obese mice. As shown in Figure 10D, HFD decreased the phosphorylation of AMPK and ACC in the liver, which was reversed by PCW administration.

### 2.9. Poricoic Acid, Pachymic Acid, and Ergosterol are Responsible for PCW-Mediated Attenuation of Hepatic Steatosis

The main bioactive components in PCW include polysaccharides, triterpenoids, fatty acids, sterols, and enzymes [11]. To identify which compounds are responsible for PCW-mediated amelioration of hepatic steatosis, we selected three major chemical components of PCW—poricoic acid, pachymic acid, and ergosterol, and evaluated their effects on hepatic steatosis in HepG2 cells treated with FFA. To determine the concentration of each compound used in the study, we first examined the cytotoxicity of each compound on HepG2 cells. As shown in Figure 11A, cell viability was not affected by up to 12.5 μM of poricoid acid, 1.25 μM of pachymic acid, and 1.25 μM of ergosterol, respectively. Thus, the beneficial effects of these three compounds on hepatic steatosis were investigated at 6.25 μM and 12.5 μM for poricoid acid, and 0.63 μM and 1.25 μM for both pachymic acid and ergosterol, in HepG2 cells treated with FFA. As shown in Figure 11B, western blot analysis revealed that all three compounds directly induced the phosphorylation of AMPK and ACC in HepG2 cells. We then assessed the inhibitory effects of these three compounds on intracellular TG accumulation in HepG2 cells treated with FFA. As shown in Figure 11C, all of three compounds significantly decreased FFA-induced intracellular TG accumulation consistent with an increase in AMPK phosphorylation. These results indicated that poricoic acid, pachymic acid, and ergosterol are responsible for PCW-mediated attenuation of hepatic steatosis.

## 3. Discussion

Hepatic steatosis is the first step in the development of NAFLD; therefore, development of compounds that can alleviate hepatic steatosis can potentially help in treating NAFLD. PCW, a mushroom with bioactive compounds, has been widely used as herbal medicine in Asia given its diverse pharmacological effects including anti-cancer and anti-inflammation properties [11]. However, its anti-hepatic steatosis effects have not been reported. We, therefore, investigated if PCW extracts exerted any therapeutic activity against hepatic steatosis in HepG2 cells treated with FFA and in HFD obese mice. Our results showed that PCW significantly reduced hepatic TG accumulation in both FFA-treated HepG2 cells and HFD obese mice, suggesting that PCW exerts protective activity against hepatic steatosis. Furthermore, we observed that PCW increased the phosphorylation of AMPK, ACC, and SREBP1c in HepG2 cells treated with FFA and inhibited de novo lipogenesis. PCW reduced ER stress, but stimulated fatty acid oxidation and autophagy in HepG2 cells treated with FFA. However, inhibition of AMPK significantly reversed these beneficial effects, suggesting that PCW ameliorates hepatic steatosis through AMPK activation.

AMPK activation inhibits hepatic steatosis through multiple mechanisms [5,6,7]. First, AMPK inhibits lipogenesis by inactivating ACC and SREBP1c. ACC is a lipogenesis enzyme and regulates the synthesis of malonyl-CoA. AMPK inactivates ACC via phosphorylation. Inactivation of ACC by phosphorylation leads to decreased malonyl-CoA levels, thereby inhibiting hepatic lipogenesis and stimulating fatty acid oxidation. Our findings indicated that PE stimulates the phosphorylation of AMPK and ACC in HepG2 cells, and that PE inhibits lipogenesis and caused reduced accumulation of TG via AMPK-medicated phosphorylation of ACC. Furthermore, AMPK activation inhibits the transcriptional activity of SREBP1c through the phosphorylation of SREBP1c. The phosphorylation of SREBP1c at Ser372 inhibits proteolytic processing and translocational activation of SREBP-1, and subsequently suppresses the expression of SREBP1c target lipogenesis genes including *FAS*, *ACC*, and *SCD1* as well as that of *SREBP1c*. Our data revealed that PCW enhances the phosphorylation of SREBP-1c (Ser372) concomitantly with increased phosphorylation of AMPK, and decreased expression of lipogenesis genes including *SREBP1c, FAS, ACC*, and *SCD1* in HepG2 cells, suggesting that PCW extract also inhibits lipogenesis through phosphorylation and inactivation of SREBP1c. However, treatment with an AMPK inhibitor significantly reversed PCW-mediated downregulation of lipogenesis genes, demonstrating that the beneficial effect of PCW on lipogenesis is associated with AMPK activation.

Secondly, AMPK stimulates fatty acid oxidation through upregulation of PPARα-mediated fatty acid oxidation genes, including *CPT1* and *ACO* [5]. Our results revealed that FFA treatment downregulates the expression of *PPARα* and its target fatty acid oxidation genes in HepG2, whereas pre-treatment with PCW reversed such effects. Inhibition of AMPK using an AMPK inhibitor significantly blocked PCW-induced upregulation of fatty acid oxidation genes, indicating that PCW-stimulated fatty acid oxidation is dependent on AMPK activation, which may also contribute to an anti-hepatic steatosis action.

Thirdly, AMPK activation inhibits ER stress induction resulting in attenuation of hepatic steatosis [6]. Long-term activation or unresolved ER stress results in hepatic lipid accumulation through increased lipogenesis via SREBP1c activation and decreased export of intracellular TG outside via VLDL [8]. Pharmacological modulators of ER stress could, therefore, be therapeutic candidates for NAFLD [8]. In this study, we investigated whether PCW inhibits FFA-induced ER stress in HepG2 cells. We observed that PCW significantly prevents the induction of ER stress in HepG2 cells treated with FFA. Consistent with the attenuation of ER stress, our results demonstrated that PCW inhibits lipogenesis and represents a mechanism by which ER induces lipid accumulation. The inhibition of PCW-mediated attenuation of ER stress was significantly blocked by AMPK inhibition, suggesting that PCW-mediated attenuation of ER is dependent on AMPK activation. Activated AMPK exerts a protective effect on ER stress through several pathways [6,14,15]. One of them is mediated through inhibition of mTOR-dependent signaling [6]. Excess nutrients activate mTORC1 signaling, which induces ER stress and the expression of *SREBP1c* and lipogenesis genes, leading to subsequent hepatic TG accumulation. Thus, the inhibition of mTORC1 by AMPK activation prevents excess nutrient-induced hepatic lipid accumulation via suppression of ER stress and SREBP1c-dependent lipogenesis. Our results revealed that PCW reduced the FFA-induced phosphorylation of mTOR and P70S6k, and the expression of SREBP1c and its target lipogenesis genes, consistent with increased phosphorylation of AMPK. This suggested that PCW inhibits ER stress by inhibiting mTORC1-dependent lipogenesis via AMPK activation. Another mechanism involved in the inhibition of ER stress by AMPK is through the expression of oxygen-regulated protein (ORP150), an ER-associated chaperon, and sarcoendoplasmic reticulum Ca^2+^-ATPase 2b (SERCA2b), which inhibits ER stress [14,15]. ORP150 plays a protective role in ER stress by increasing the capacity of ER for protein folding and degradation, consequently ameliorating ER stress [16]. AMPK activation increases ER stress capacity for protein folding and degradation by overexpression of ORP150, which leads to inhibition of ER stress. Furthermore, SERCA is an ER membrane protein that plays a role in the uptake of Ca^2+^ from the cytosol to the ER lumen. A dysfunctional SERCA leads to Ca^2+^ release from the ER lumen, and results in alteration of ER homeostasis and subsequently the ER stress [17]. Upregulation of SERCA improves altered calcium balance during ER stress, which leads to attenuation of ER stress. To characterize the detailed mechanism by which PCW inhibits ER stress, further study involving effects of PCW on the expression of *ORP150* and *SERCA2b* is needed.

Fourthly, AMPK activation exerts a positive effect on autophagy [7]. Autophagy plays an important role in intracellular lipid degradation by mobilizing lipid droplets after the formation of the autophagosome to the lysosome for degradation of the lipid droplets (LD) by acid hydrolase (lipophagy) [9]. The inhibition of autophagy-mediated degradation of intracellular TG plays an essential role in the development of hepatic steatosis. Hepatic lipophagy was significantly decreased in obese and NAFLD model mice [18]. Therefore, pharmacological activation of lipophagy could be a potential therapeutic target against hepatic steatosis. Our study revealed that PCW reverses FFA-mediated inhibition of autophagosome formation-related proteins including LC3A/B, Beclin, ATG 3, 3, 16, in HepG2 cells treated with FFA, indicating that PCW also exhibits anti-hepatic steatosis effects through activation of lipophagy. AMPK activates lipophagy through the inhibition of mTORC1 signaling, which inhibits autophagy by suppressing autophagosome formation. AMPK activation decreases phosphorylation of mTOR and p70S6K, downstream of mTOR, resulting in efficient autophagosome formation and lysosome biogenesis. Our results indicated that PCW reduces FFA-induced phosphorylation of mTOR and p70S6K concomitantly increases AMPK phosphorylation; however, treatment with compound C significantly blocks these effects, suggesting that PCW activates autophagy by inhibiting mTOR via AMPK, which may also contribute to attenuate hepatic steatosis. Autophagy and ER stress regulate each other [19]. Based on our results, PCW-mediated suppression of ER stress may contribute to the activation of autophagy by PCW. In contrast, activation of autophagy plays a role in PCW-mediated inhibition of ER stress. Taken together, our results indicated that activation of AMPK/mTOR-dependent autophagy pathway plays an important role in PCW-mediated protection of hepatic steatosis.

The protective effects of PCW against hepatic steatosis in HFD-induced obese mice were further confirmed in vivo. HFD-induced hepatic steatosis, characterized by increased hepatic lipid accumulation, was efficiently blocked by PCW administration, as seen from the histological examination and hepatic TG measurements. PCW also restored HFD-mediated reduction of AMPK phosphorylation. These results confirmed that PCW exhibits anti-hepatic steatosis effects via AMPK activation in vivo.

To determine which components in PCW are responsible for PCW-mediated amelioration of hepatic steatosis, we investigated the beneficial effects of three major chemical components present in PCW—poricoic acid, pachymic acid, and ergosterol in HepG2 cells treated with FFA. All three compounds exhibited anti-hepatic steatosis effects consistent with activation of AMPK in HepG2 cells treated with FFA. Further studies are necessary to ascertain the anti-hepatic steatosis effects in vivo and to characterize the underlying mechanisms.

In conclusion, PCW exerts a protective effect against hepatic steatosis in HepG2 cells treated with FFA and HFD obese mice through the regulation of lipid metabolism, inhibition of ER stress, and activation of autophagy in an AMPK-dependent manner. PCW extract has therapeutic potential for the treatment of NAFLD. Further studies are needed to verify the beneficial effects of PCW, particularly in reducing the risk of other metabolic disorders.

## 4. Materials and Methods

### 4.1. Reagents

Palmitate, oleate, and compound C were purchased from Sigma-Aldrich (St. Louis, MO, USA). Antibodies against AMPKα, phospho-AMPKα (Thr172), SREBP1c, FAS, PPARα, GRP78, CHOP, XBP1C, p-PERK (Thr981), PERK, and actin were purchased from Santa Cruz Biotechnology (Santa Cruz, CA, USA). Antibodies against ACC, phospho-ACC (Ser79), phosphor-SREBP1c (Ser372), LC3A/B, Beclin1, ATG3, ATG7, ATG16L1, phospho-mTOR (Ser2448), mTOR, phospho-P70S6K (Thr389), and P70S6K were purchased from Cell Signaling Technology (Danvers, MA, USA). SREBP–luciferase (Luc) and FAS–Luc reporters were purchased from Addgene (Cambridge, MA, USA).

### 4.2. Preparation of PCW Extract

The mushroom was identified by Dr. Yun Tai Kim based on the “Illustrated Guide to Clinical Medical Herbs” [20], and a voucher specimen (#NP-1090) was deposited at the Research group of functional food materials, Korea Food Research Institute. Dried PCW (900 g) was extracted with 95% ethanol (9000 mL) for 6 h at 80 °C. The extract was filtered through a membrane filter (0.45 µm; Millipore, Billerica, MA, USA). After removing the solvent via rotary evaporation, the remaining extract was freeze-dried, yielding about 1.25% of the dried weight (*w*/*w*).

### 4.3. Cell Culture

HepG2 cells purchased from the American Type Culture Collection (ATCC, CL-173TM; Manassas, VA, USA) were maintained in Dulbecco’s modified-Eagle’s medium (DMEM; HyClone, Logan, Utah, USA) containing 10% fetal calf serum (FCS; HyClone), 50 units/mL penicillin, and 50 mg/mL streptomycin at 37 °C in a humidified atmosphere with 5% CO2. HepG2 cells were treated with FFA mixture (palmitate and oleate, 1:2 ratio) in the presence or absence of PCW for 24 h to measure protein levels or TG accumulation.

### 4.4. Cytotoxicity Assay

Cytotoxicity of FFA and/or PCW was measured using the 3-(4,5-dimethylthiazol-2-yl)-2,5-diphenyltetrazolium bromide (MTT) assay. Briefly, cells were seeded in 96-well plates at a density of 1 × 104 cells per well and treated with various concentrations of FFA (0, 0.25, 0.5, 1, and 2 mM) and/or PCW (0, 20, 40, 80, and 160 μg/mL). After incubating for 24 h, MTT (1 mg/mL) was added and the plate was further incubated for 4 h at 37 °C. The absorbance was measured at 570 nm using a microplate reader.

### 4.5. ORO Staining

HepG2 cells were washed twice with phosphate-buffered saline (PBS) and fixed with 10% formaldehyde for 30 min at room temperature. After fixation, the cells were washed again with PBS and stained with an ORO working solution (1.5 mg/mL ORO/60% isopropanol) and incubated for 30 min at room temperature. The cells were then rinsed with 75% ethanol to remove unbound dye, and were subsequently washed with PBS prior to being imaged using a light microscope. The dye was dissolved using 100% isopropanol and absorbance was measured at 510 nm using a microplate reader.

### 4.6. TG Measurement

HepG2 cell suspensions and mouse liver lysates were mixed with 750 μL of chloroform/methanol/H2O (8:4:3, *v*/*v*/*v*) to extract the TGs. The cell suspensions or liver lysates were incubated at room temperature for 1 h and centrifuged at 800 ×g for 10 min. The bottom layer (organic phase) obtained was dried overnight and dissolved in ethanol prior to determining the TG concentration using an enzyme reaction kit (Asan Pharmaceutical, Seoul, Republic of Korea), and the values were normalized to the protein content.

### 4.7. Western Blot

HepG2 cells and liver tissue were lysed in ice-cold lysis buffer containing the protease inhibitor cocktail and 1 mM phenylmethanesulfonyl fluoride for 30 min, and were subjected to centrifugation at 10,000× *g* for 30 min at 4 °C. Proteins (50 μg) were subjected to sodium dodecyl sulfate polyacrylamide gel electrophoresis, and were transferred onto polyvinylidene difluoride membranes (Millipore, Billerica, MA, USA). The membranes were blocked using 5% non-fat skimmed milk for 30 min at room temperature and were probed with primary antibodies. After washing with Tween 20/Tris-buffered saline (T-TBS), the membranes were incubated with horseradish peroxidase-conjugated secondary antibody (1:1000) for 1 h at room temperature. Membranes were then washed three times with T-TBS and the proteins were detected using an enhanced chemiluminescence (ECL) western blot detection kit (Amersham, Uppsala, Sweden).

### 4.8. qPCR

Total RNA was isolated from HepG2 cells using TRIzol^®^ (Ambion Invitrogen, Carlsbad, CA, USA). cDNA was generated from 1 μg of Total RNA using the GoScript™ reverse transcription system (Promega, USA) as per the manufacturer’s protocol. PCR amplification was performed using SYBR Green premixed Taq reaction mixture and gene-specific primers. The primer sequences used in this study were: *SREBP1c* 5′-CGGAGCCATGGATTGCACT-3′ (sense) and 5′-TAGGCCAGGGAAGTCACTG-3′ (antisense); *FAS* 5′-TCGTGGGCTACAGCATGGT-3′ (sense) and 5′-GCCCTCTGAAGTCGAAGAAGAA-3′ (antisense); *SCD1* 5′-CCAGTCAACTCCTCGCACTT (sense) and 5′-AGCCAGGTTTGTAGTACCTCC-3′ (antisense); *ACC* 5′-CTGTAGAAACCCGGACAGTAGAAC-3′ (sense) and 5′-GGTCAGCATACATCTCCATGTG-3′ (antisense); *PPARα* 5′-GGACAGCAAATCTTGAAGCAGC-3′ (sense) and 5′-CTCTGATCCCTCTAGCACCTT-3′ (antisense); *CPT1* 5′-TTTCCTTGCTGAGGTGCTCT-3′ (sense) and 5′-TCTCGCCTGCAATCATGTAG-3′ (antisense); and *ACO* 5′-CAGGAA AGTTGGTGTGTGGC-3′ (sense) and 5′-AATCTGGCTGCACGGAGTTT-3′ (antisense).

### 4.9. Luciferase Assay

HepG2 cell were transiently transfected with SREBP1c–Luc reporter plasmid or FAS–Luc reporter plasmid and incubated for 24 h. The cells were then treated with FFA (1 mM) and/or PCW (20 or 40 μg/mL) for 24 h. Luciferase activity was measured using an Luc assay system (Promega, Madison, WI, USA).

### 4.10. Animal Experiments

C57BL/6 mice (male, 6 weeks old) were obtained from Jung-Ang Lab Animal, Inc. (Seoul, Korea). The animals were housed under optimal temperature (21–23 °C) and humidity (40–60%) conditions, with a 12 h light/dark cycle, and were given free access to food and water. The mice were fed a normal diet (ND) or an HFD for 6 weeks. Next, the HFD-fed mice were randomly divided into the following three groups (*n* = 8 per group): HFD (distilled water-treated) group, HFD + low-dose PCW (100 mg/kg of body weight) group, and HFD + high-dose PCW (300 mg/kg of body weight) group. The experimental diet used was TD.06414, based on HFD containing 60% kcal fat. The control diet contained 10% kcal fat. PCW was administered orally daily for 6 weeks. The animal experiment protocol used in this study was reviewed and approved by Pusan National University’s Institutional Animal Care and Use Committee, in accordance with the established ethical and scientific care procedures (Approval No. PNU-2018-1832, 12/01/2018).

### 4.11. Statistical Analysis

All data are presented as mean ± SD values. Data were analyzed using one-way ANOVA, and the differences between means were determined using Tukey-Kramer post-hoc test. Values were considered statistically significant at *p* < 0.05.

## Figures and Tables

**Figure 1 ijms-20-04801-f001:**
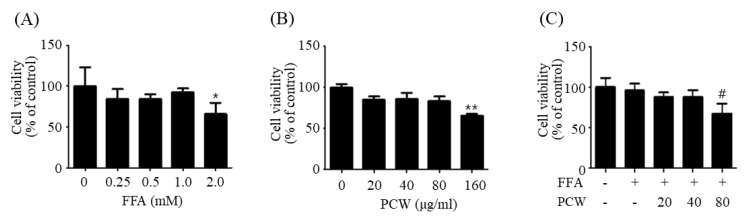
Cytotoxicity of free fatty acid (FFA) and *Poria cocos* Wolf (PCW) on HepG2 cells. (**A**) HepG2 cells were treated with various concentrations of FFA (0, 0.25, 0.5, 1.0, or 2.0 mM) for 24 h. (**B**) HepG2 cells were treated with various concentrations of PCW (0, 20, 40, 80, or 160 μg/mL) for 24 h. (**C**) HepG2 cells were co-treated with FFA (1 mM) and PCW (20, 40, or 80 μg/mL) for 24 h. Cell viability was determined by 3-(4,5-dimethylthiazol-2-yl)-2,5-diphenyltetrazolium bromide (MTT) assay. Data represent mean ± SD of triplicate experiments. * *p* < 0.05 and ** *p* < 0.01 versus untreated control. # *p* < 0.05 versus FFA-treated cells.

**Figure 2 ijms-20-04801-f002:**
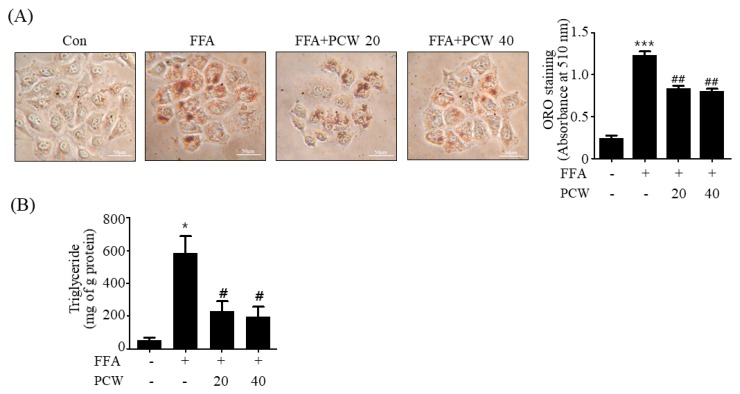
PCW inhibits FFA-induced triglyceride (TG) accumulation in HepG2 cells. HepG2 cells were treated with FFA (1 mM) and/or PCW (20 or 40 μg/mL) for 24 h. (**A**) Oil Red O (ORO) staining was performed and visualized under a light microscope (200×). Bars in the graphs represent intracellular lipid content determined by ORO staining. (**B**) Measurement of intracellular TG amount. Data represent mean ± SD of triplicate experiments. * *p* < 0.05 and *** *p* < 0.001 versus untreated control. # *p* < 0.05, ## *p* < 0.01 versus FFA-treated cells.

**Figure 3 ijms-20-04801-f003:**
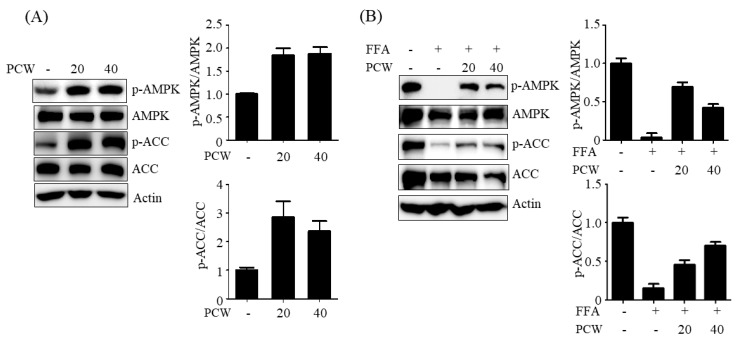
PCW activates the AMP-activated protein kinase (AMPK) pathway in HepG2 cells treated with FFA. (**A**) HepG2 cells were incubated with PCW (20 and 40 μg/mL) for 24 h. Phosphorylation of AMPK/acetyl-CoA carboxylase (ACC) was determined by the western blot. (**B**) HepG2 cells were treated with FFA (1 mM) and/or PCW (20 or 40 μg/mL) for 24 h. Phosphorylation of AMPK/ACC was determined by the western blot. Bar graphs represent densitometric analyses of band intensity ratios of p-AMPK/AMPK and p-ACC/ACC.

**Figure 4 ijms-20-04801-f004:**
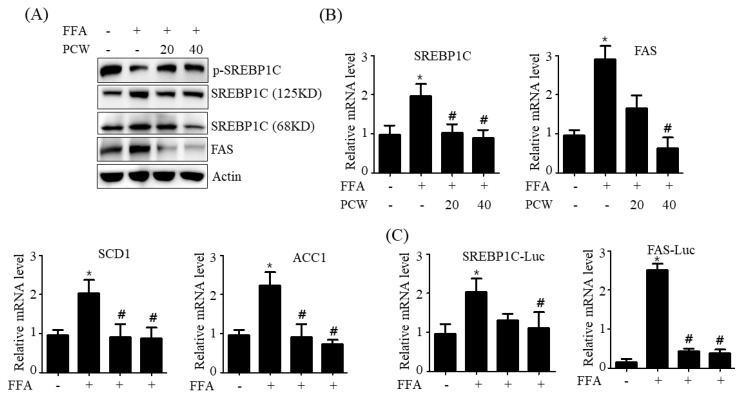
PCW inhibits sterol regulatory element-binding protein 1c (SREBP1c)-mediated lipogenesis in HepG2 cells treated with FFA. HepG2 cells were treated with FFA (1 mM) and/or PCW (20 and 40 μg/mL) for 24 h. (**A**) Protein levels of p-SREBP1c, SREBP1c, and FAS were analyzed by the western blot. Densitometric analysis of band intensity is shown in Appendix A (**B**) Relative mRNA levels of *SREBP1c* and its target lipogenesis genes were assayed by qPCR. (**C**) HepG2 cells were transfected with the SREBP1c–luciferase (Luc) reporter or FAS–Luc reporter and incubated for 24 h. The transfected cells were further treated with FFA (1 mM) and/or PCW (20 and 40 μg/mL) for additional 24 h. Luc activity was determined. Data represent mean ± SD of triplicate experiments. * *p* < 0.05 versus untreated control. # *p* < 0.05 versus FFA-treated cells.

**Figure 5 ijms-20-04801-f005:**
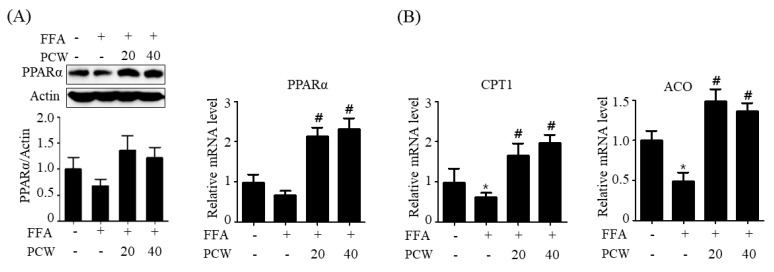
PCW stimulates the expression of fatty acid oxidation genes in HepG2 cells treated with FFA. HepG2 cells were treated with FFA (1 mM) and/or PCW (20 or 40 μg/mL) for 24 h. (**A**) Protein levels of PPARα were analyzed by the western blot. Bar graphs represent densitometric analysis of the band intensity ratios of PPARα/Actin. Relative mRNA levels of *PPARα* were assayed by qPCR. (**B**) Relative mRNA levels of carnitine palmitoyltransferase (*CPT1*) and acyl-coenzyme A oxidase (*ACO*) were assayed by qPCR. Data represent mean ± SD of triplicate experiments. * *p* < 0.05 versus untreated control. # *p* < 0.05 versus FFA-treated cells.

**Figure 6 ijms-20-04801-f006:**
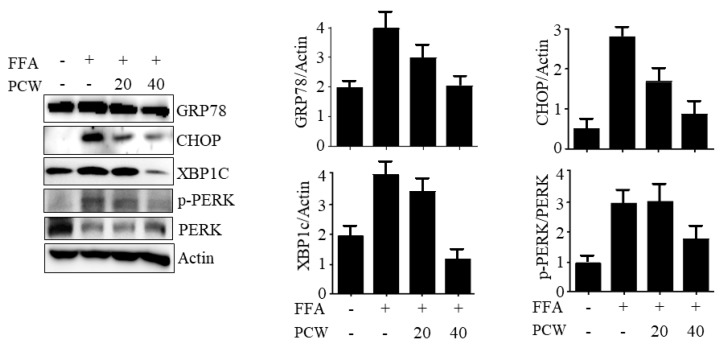
PCW alleviates endoplasmic reticulum (ER) stress in HepG2 cells treated with FFA. HepG2 cells were treated with FFA (1 mM) and/or PCW (20 or 40 μg/mL) for 24 h. Protein levels of ER stress markers were analyzed by the western blot. Bar graphs represent densitometric analysis of band intensity ratios of GRP78/Actin, CHOP/Actin, XPB1c/Actin, and p-PERK/PERK.

**Figure 7 ijms-20-04801-f007:**
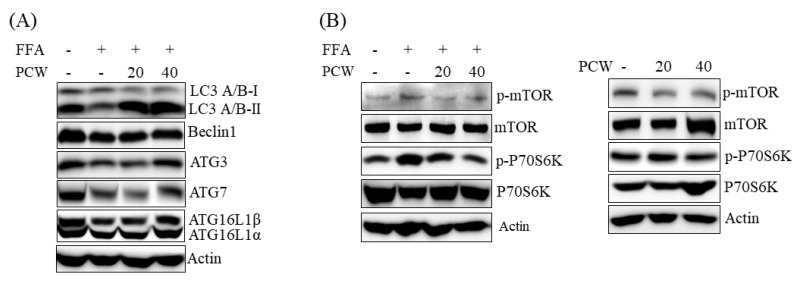
PCW activates autophagy in HepG2 cells treated with FFA. HepG2 cells were treated with FFA (1 mM) and/or PCW (20 or 40 μg/mL) for 24 h. (**A**) Protein levels of autophagy markers were analyzed by the western blot. (**B**) The phosphorylation of p-mTOR/p-P79S6K was determined by the western blot. Densitometric analysis of band intensity is shown in Appendix A.

**Figure 8 ijms-20-04801-f008:**
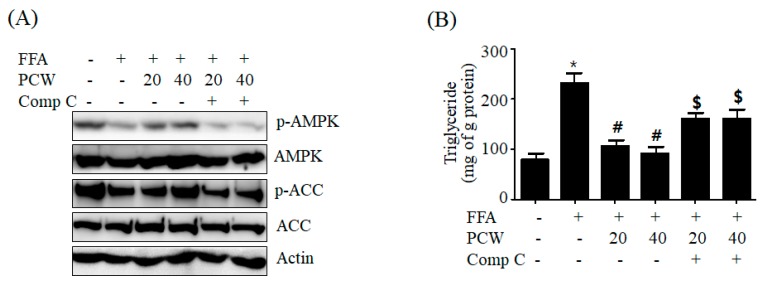
Inhibition of AMPK prevents PCW-mediated reduction of intracellular TG accumulation in HepG2 cells treated with FFA. HepG2 Cells were pre-treated with compound C (Comp C, 10 μM) for 3 h and then treated with FFA (1 mM) and PCW (20 or 40 μg/mL) for 24 h. (**A**) The phosphorylation of AMPK/ACC was determined by the western blot. Densitometric analysis of band intensity is shown in Appendix A. (**B**) Measurement of intracellular TG. Data represent mean ± SD of triplicate experiments. * *p* < 0.05 versus untreated control. # *p* < 0.05 versus FFA-treated cells. $ *p* < 0.05 versus FFA- and PCW-treated cells.

**Figure 9 ijms-20-04801-f009:**
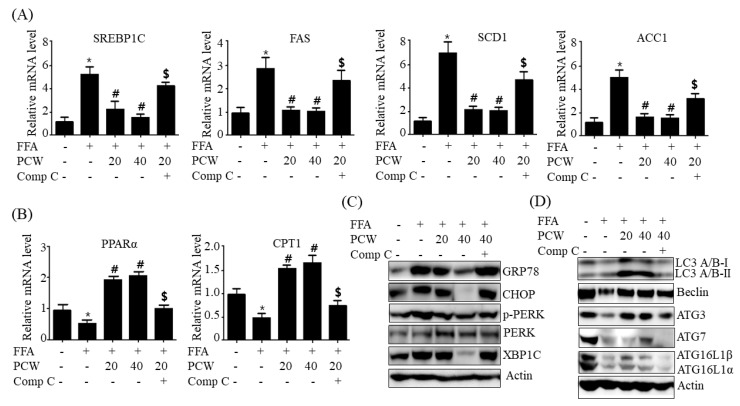
Inhibition of AMPK using compound C reverses PCW-mediated effects on the expression of lipogenesis and fatty acid oxidation genes, ER stress, and autophagy in HepG2 cells treated with FFA. HepG2 Cells were pre-treated with compound C (Comp C, 10 μM) for 3 h and then treated with FFA (1 mM) and PCW (20 or 40 μg/mL) for 24 h. (**A**) Relative mRNA levels of *SREBP1C*, *FAS*, *SCD1*, and *ACC1* assayed by qPCR. (**B**) Relative mRNA levels of *PPAR*α and *CPT1* were assayed by qPCR. (**C**) Protein levels of ER stress markers were analyzed by the western blot. Densitometric analysis of band intensity is shown in Appendix A. (**D**) Protein levels of autophagy proteins were analyzed by the western blot. Densitometric analysis of band intensity is shown in Appendix A. Data represent mean ± SD of triplicate experiments. * *p* < 0.05 versus untreated control. # *p* < 0.05 versus FFA-treated cells. $ *p* < 0.05 versus FFA- and PCW-treated cells.

**Figure 10 ijms-20-04801-f010:**
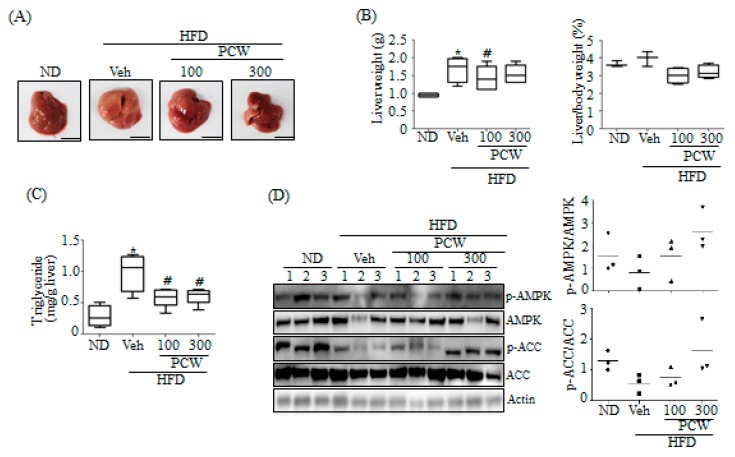
PCW ameliorates high-fat diet (HFD)-induced hepatic steatosis through AMPK activation. C57BL6 mice were fed HFD for 6 weeks, and then PCW was administered orally for another 6 weeks. (**A**) Representative images of liver morphology. A scale bar is 1 cm. (**B**) Liver weight and ratio of liver to body weight. Data represent mean ± SD of 5 mice. * *p* < 0.05 versus ND (normal diet) mice. # *p* < 0.05 versus HFD mice. (**C**) Measurement of hepatic TG levels. Data represent mean ± SD of 5 mice. * *p* < 0.05 versus ND mice. # *p* < 0.05 versus HFD mice. (**D**) Phosphorylation of AMPK/ACC was determined in liver lysates by the western blot.

**Figure 11 ijms-20-04801-f011:**
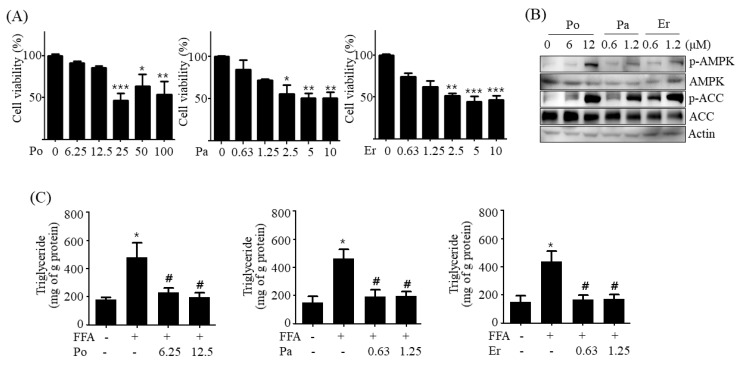
Poricoic acid, pachymic acid, and ergosterol are responsible for PCW-mediated attenuation of hepatic steatosis. (**A**) HepG2 cells were treated with various concentrations of poricoic acid (Po) (0, 6.25, 12.5, 25, 50, or 100 μM), pachymic acid (Pa) (0, 0.63, 1.25, 2.5, 5, or 10 μM) or ergosterol (Er) (0, 0.63, 1.25, 2.5, 5, or 10 μM) for 24 h. Cell viability was determined by MTT assay. (**B**) HepG2 cells were treated with Po (6 or 12 μM), Pa (0.6 or 1.2 μM), or Er (0.6 or 1.2 μM) for 24 h. The phosphorylation of AMPK/ACC was determined by the western blot. Densitometric analyses of band intensity ratios for p-AMPK/AMPK and p-ACC/ACC are shown in Appendix A (Appendix A). (**C**) Intracellular TG amounts were measured. Data represent mean ± SD of triplicate experiments. * *p* < 0.05, ** *p* < 0.01, and *** *p* < 0.001 versus untreated control. # *p* < 0.05 versus FFA-treated cells.

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
