# Peer review of "Poria cocus Wolf Extract Ameliorates Hepatic Steatosis through Regulation of Lipid Metabolism, Inhibition of ER Stress, and Activation of Autophagy via AMPK Activation"

_ijms, 2019, doi:10.3390/ijms20194801_

Round 1

Reviewer 1 Report

The manuscript is written clearly and comprehensible. It describes potential molecular mechanism of the protective effects of  Poria cocus  mushroom extract on lipid metabolism and against non-alcoholic fatty liver disease in the in vitro (HepG2 cells treated with free fatty acids) and in vivo (C57BL/6 mice fed high-fat diet) using Western blot, qPCR and Luciferase assay to assess genes and proteins involved with lipid metabolism.

The area of the investigation presented in this paper is interesting and associated with new trends in pharmacological and molecular studies.  The research described in this paper is well-designed and conducted and, moreover, the analysis of the results is properly developed. The  Authors show practical implications of obtained results and indicate that Poria cocus extracts affect lipid metabolism and have a protective effect against hepatic steatosis in HepG2 cells as well in obese mice. According to these findings – Poria cocus extracts supplementation may, to some extent, prevent non-alcoholic fatty liver disease.

Minor revision:

Please use Western blot instead of western blot in Figures description. In the Method section, “Preparation of PCW extract” please change: The mushroom was identified… (Mushrooms are not plants). Are the Authors planning to use standardized mycelium extracts from in vitro culture? Fruiting bodies can have variable content of secondary metabolites and bioelements depending on the ground.

Author Response

The manuscript is written clearly and comprehensible. It describes potential molecular mechanism of the protective effects of Poria cocus mushroom extract on lipid metabolism and against non-alcoholic fatty liver disease in the in vitro (HepG2 cells treated with free fatty acids) and in vivo (C57BL/6 mice fed high-fat diet) using Western blot, qPCR and Luciferase assay to assess genes and proteins involved with lipid metabolism.

The area of the investigation presented in this paper is interesting and associated with new trends in pharmacological and molecular studies. The research described in this paper is well-designed and conducted and, moreover, the analysis of the results is properly developed. The Authors show practical implications of obtained results and indicate that Poria cocus extracts affect lipid metabolism and have a protective effect against hepatic steatosis in HepG2 cells as well in obese mice. According to these findings – Poria cocus extracts supplementation may, to some extent, prevent non-alcoholic fatty liver disease.

Minor revision:
Please use Western blot instead of western blot in Figures description. In the Method section, “Preparation of PCW extract” please change: The mushroom was identified… (Mushrooms are not plants). Are the Authors planning to use standardized mycelium extracts from
in vitro culture? Fruiting bodies can have variable content of secondary metabolites and bioelements depending on the ground.

Response;
We appreciate your valuable comments and suggestions. As you suggested, we revised western blot in Figures description to “Western blot” and changed plant to “mushroom”.
Further, as you suggested, we will try to use standardized mycelium extracts at next experiments.

Reviewer 2 Report

Kim et al have investigated the anti-steatotic effects of PCW both in vitro and in vivo. They found that PCW attenuates an increase of triglyceride levels induced by free fatty acids, most likely due to activation of AMPK. Furthermore, PCW had beneficial effects on ER stress and autophagy. The paper was well-structured and it was easy to understand the results presented. However, there are some questions that should be answered before it’s acceptable for publication.

The conclusions rely heavily on expression analysis data. Although this gives some insight into the mechanisms behind the observed effects, additional functional testing should be conducted. Assays like beta-oxidation, free fatty acid uptake and triglyceride secretion should be performed to prove that the expression changes are of importance.

The use of oleate in cell culture experiments may not have been the best choice. Oleate is generally considered protective against lipotoxic stress.

Figure 2A clearly shows less cells in FFA and PCW treated wells, although the authors show that the viability is not changed under these conditions. How is this discrepancy explained? Also, it is unclear to me what the values on the y-axes for ORO staining are. What is the unit?

ORO stain is known to stick to plastic, how did the authors make sure that there was no unspecific binding in their ORO measurments?

Throughout the paper, it is unclear how many wells were included for each independent experiment. Please clarify this.

In the bar diagram in figure 3A, FFA is included, although the legend and the blots shown say it is not included. Please clarify.

In line 152, the authors write “treatment with FFA significantly reduced the phosphorylation of AMPK and ACC in HepG2 cells”. But figure 3B doesn’t contain anything to indicate any significance. If it is in fact not significant, please revise the sentence.

The headline on line 162, “PCW inhibits lipogenesis and stimulates fatty acid oxidation in HepG2 cells treated with FFA”, is too strongly worded. More correct would be to write “PCW inhibits markers of lipogenesis…”.

Figure 5A is missing labels on top of the blot. Also, the increase of PPARa expression after PCW treatment seem to be significantly higher than basal. If so, please indicate.

Throughout the figures, there must be a mistake with the plus signs indicating FFA treatment, as all bars have a plus underneath, when I assume the first bar is always without any treatment at all.

All immunoblots should be quantified if conclusions are to be made about the increase/decrease. Showing just blots without any numbers are not enough.

The pathophysiology of fatty liver and its progression is influenced by multiple factors in a “multiple parallel-hit model,” in which oxidative stress plays a very likely primary role as the starting point of the hepatic and extrahepatic damage. What is the effect of PCW on oxidative stress?

In figure 6 there are no error bars. Are these results just based on one experiment? If so, please increase to 3 independent experiments, as has been done for the other cell experiments.

For some measurements, the effect of PCW alone has been evaluated (e.g. blots in figure 3A, right panel in figure 7B). What was the rationale for not measuring the effect of PCW for all investigations performed?

In figure 9, how come only PCW 20, and not PCW 40, is included when combined with compound C?

In figure 9C, blots for PCW 20 in combination with compound C seem to be in the same range as PCW 20 alone. Therefore, the authors can not claim that the effect seen in ER stress is AMPK-dependent.

Figure 7B (left panel) and 10D are missing loading controls.

The authors describe 8 mice being used per group in the animal experiments, but not all mice are included in the analyses (n=5 for liver weight and triglyceride measurement, n=3 for western blot). Why are not all mice included?

Round 2

Reviewer 2 Report

OK!